# Aryl Hydrocarbon Receptor (AHR) Ligands as Selective AHR Modulators (SAhRMs)

**DOI:** 10.3390/ijms21186654

**Published:** 2020-09-11

**Authors:** Stephen Safe, Un-ho Jin, Hyejin Park, Robert S. Chapkin, Arul Jayaraman

**Affiliations:** 1Department of Veterinary Physiology and Pharmacology, Texas A&M University, College Station, TX 77843, USA; jinunho@gmail.com (U.-h.J.); hpark@cvm.tamu.edu (H.P.); 2Departments of Nutrition and Food Science and Biochemistry and Biophysics, Texas A&M University, College Station, TX 77843, USA; r-chapkin@tamu.edu; 3Department of Chemical Engineering, Texas A&M University, College Station, TX 77843, USA; arulj@mail.che.tamu.edu

**Keywords:** AhR, ligand selectivity, agonist, antagonist

## Abstract

The aryl hydrocarbon receptor (AhR) was first identified as the intracellular protein that bound and mediated the toxic effects of 2,3,7,8-tetrachlorodibenzo-p-dioxin (TCDD, dioxin) and dioxin-like compounds (DLCs). Subsequent studies show that the AhR plays an important role in maintaining cellular homeostasis and in pathophysiology, and there is increasing evidence that the AhR is an important drug target. The AhR binds structurally diverse compounds, including pharmaceuticals, phytochemicals and endogenous biochemicals, some of which may serve as endogenous ligands. Classification of DLCs and non-DLCs based on their persistence (metabolism), toxicities, binding to wild-type/mutant AhR and structural similarities have been reported. This review provides data suggesting that ligands for the AhR are selective AhR modulators (SAhRMs) that exhibit tissue/cell-specific AhR agonist and antagonist activities, and that their functional diversity is similar to selective receptor modulators that target steroid hormone and other nuclear receptors.

## 1. Introduction

Intracellular receptors such as the nuclear receptor (NR) superfamily play important roles in maintaining cellular homeostasis and in pathophysiology [1,2]. Members of the steroid hormone receptor subfamily were among the first NR’s identified, and the binding and effects of their endogenous hormonal ligands such as 17β-estradiol have been extensively investigated and form a conceptual basis for understanding hormone-hormone receptor action [3]. The aryl hydrocarbon receptor (AhR) is also an intracellular receptor that is a member of the basic-helix-loop-helix (bHLH) family of genes that exhibit a mode of action similar to that described for many NRs [4,5,6]. The cytosolic AhR is bound to several factors, including hsp90, AIP/XAP2 and p23, and treatment with an AhR ligand results in nuclear uptake of the bound receptor complex that forms a nuclear heterodimer with the AhR nuclear translocator (Arnt) protein. This heterodimer interacts with cis-dioxin/xenobiotic response elements (DRE/XRE), which have a core GCGTG pentanucleotide sequence, and this results in recruitment of nuclear cofactors to activate or repress gene expression [4,5,6]. This pathway is similar to that described for several nuclear receptors such as peroxisome proliferator-activated receptors, and retinoic acid receptors, which form a heterodimer with the retinoid X receptor (RXR) [1,2]. These similarities between NRs and the AhR also extend to their increasing number of alternative mechanisms of action, which include binding of receptor complexes to non-consensus response elements; interactions with alternate partners (rather than RXR/Arnt); and acting via extranuclear regions of the cell, including the membrane and cytosol [7,8,9,10,11,12]. Despite the similarities between the AhR and steroid hormone receptors, the AhR was initially identified and characterized as the intracellular receptor not for an endogenous ligand but for 2,3,7,8-tetrachlorodibenzo-p-dioxin (TCDD), a toxic industrial and combustion by-product [13]. Some of the evidence demonstrating that the AhR protein was a receptor included studies showing that the rank order binding affinities of TCDD and structurally related dioxin-like compounds (DLCs) were similar to their rank order potencies for inducing specific toxic responses and as inducers of CYP1A1 [13,14]. Several endogenous ligands that bind the AhR have subsequently been identified, and these include 6-formyl (3,2-b) carbazole (FICZ), 2-(1′-H-indole-3-carbonyl)thiazole-4-carboxylic acid methyl ester (ITE), tryptophan metabolites such as kynurenine and other gut microbial products, and leukotrienes [15,16,17,18,19,20]. These compounds exhibit some tissue-specific modulation of AhR action; however, their role as major endogenous AhR ligands is unresolved. The differences in the initial discoveries of steroid hormone receptors and their endogenous hormonal ligands vs. the AhR receptor, and its high-affinity ligand, TCDD, has tarred the AhR as the receptor associated with toxicity and DLCs. This linkage between the AhR and toxicity has hindered the development of drugs targeting the AhR, whereas up to 13.5% of all developed drugs target NRs [21]. This review will focus on the conceptual similarities between NRs and the AhR in terms of the development of selective receptor modulators (SRMs) and their potential clinical applications in treating adverse health conditions.

## 2. AhR Functions and Potential for Drug Targeting

AhR knockout mice (AhR^−/−^) were generated by several laboratories, and initial studies confirmed that the receptor was required for mediating the toxicity of TCDD and structurally related halogenated aromatics, including polychlorinated dibenzo-p-dioxins, dibenzofurans and biphenyls [22,23,24,25,26,27]. Comparable studies with Arnt knockout mice could not be performed, since the loss of Arnt was embryonal lethal [28]. Subsequent studies demonstrated that the loss of the AhR in mice resulted in several abnormalities in multiple organs/tissues, which include liver (fibrosis, failure of developmental closure of the ductus venosis) [29]; heart (cardiac hypertrophy, elevated arterial blood pressure and increased fibrosis) [30,31,32,33,34]; multiple adverse female reproductive tract problems [35,36,37,38]; altered mammary gland development [39,40]; decreased skin barrier integrity [41]; decreased intestinal resilience (increased stem cells, increased susceptibility to inflammation, decreased barrier function immune dysfunctions and enhanced tumorigenesis) [42,43,44,45,46,47]; extensive immune dysfunctions [48,49,50,51,52,53]; modulation of stem cells [42,54,55]; enhanced formation of uric acid stones in the bladder [56]; oculomotor deficits and defective optic nerve myelin sheath [57,58]; and neuronal function deficits [59,60,61]. These results from mouse models demonstrate that although loss of the AhR in mice is not embryo-lethal, this receptor plays a role in multiple organs/tissues (Figure 1).

The potential utility of a receptor as a drug target depends on at least two critical factors, namely, that the receptor is expressed (or overexpressed) and has important functions in a diseased target tissue. There are distinct advantages in targeting a receptor. If the functional activity of the receptor enhances or inhibits the disease process then potential receptor ligand therapies include treatment with antagonists or agonists, respectively. There is extensive evidence that the AhR is expressed and functional in multiple tumors, including pancreatic, breast, lung, colon, glioma and other cancer cell lines (rev in [62,63]). For example, knockdown of the AhR in head and neck cancer cells decreases invasion and expression of growth-promoting genes such as amphiregulin, epiregulin and platelet derived growth factor A, demonstrating the pro-oncogenic activity of the AhR. Thus, treatment of these cells with an agonist (TCDD) or the AhR antagonist GNF351 enhances or inhibits, respectively, the pro-oncogenic activity of the AhR [64,65,66,67,68,69]. In intestinal cancer, the loss of the AhR enhances carcinogenesis in mouse models and treatment of wild-type mice with AhR agonists protects against chemical-induced and genetic models of colon cancer [42,43,44,45,46,47]. Studies in this laboratory also showed that the AhR exhibited tumor suppressor-like activity in glioblastoma cells [67]. This was in contrast to a previous report [68], and there are other examples of apparent differences in the functional roles of the AhR in other tumor types. The AhR plays a role in other tumors, and in some cases, the variability in the function of the AhR is due to the use of different model systems.

## 3. AhR Expression/Function in Pathophysiology

In addition to cancer, these are several examples where treatment of various diseases with AhR agonists or antagonists can be beneficial; moreover, the therapeutic targets in mouse models shows some overlap with results illustrated in Figure 1 showing tissues/organs affected by loss of the AhR. Dextran sodium sulfate (DSS)-induced colon inflammation is inhibited by multiple AhR ligands, including TCDD [69,70,71]; the inflammatory responses and genes associated with skin inflammation (Psoriasis-like) were inhibited the AhR agonist FICZ [72]. AhR agonists inhibit hepatic inflammation and fibrosis, and activation of hepatic stellate cells [73,74]; the AhR in combination with IRF4 is induced by activin A to inhibit asthma responses, and this is blocked by the AhR antagonist CH223191 [75]. The AhR is important for regulatory B cells and IL-10 production, and FICZ enhances this response to protect against arthritis [76]. The AhR antagonist GNF351 attenuates proliferative/anti-inflammatory responses in fibroblast-like synoviocytes from patients with rheumatoid arthritis [77]. The AhR also plays multiple roles in the brain [78,79], and studies showed that loss of AhR in mice resulted in impaired hippocampal-dependent contextual fear memory. Decreased neuronal differentiation was more prevalent in AhR^−/−^ than in wild-type (AhR^+/+^) mice, and in wild-type mice TCDD induced responses similar to that observed in AhR^−/−^ mice [79]. In a lung model of pulmonary inflammation known as idiopathic pneumonia syndrome (IPS), an HDAC inhibitor induced L-kynurenine and suppressed IL-17 and IL-6 expression, and these damage-related responses were reversed after treatment with the AhR antagonist CH223191 [80]. Plasma levels of the AhR agonists kynurenine and indoxyl sulfate increased in mice bearing cancer cell xenografts. This was accompanied by upregulation of TF and PAI-1 and increased clot weights in a venous thrombosis model of colon cancer [81]. The AhR antagonist CH223191 ameliorated these responses, suggesting a role for the endogenous tryptophan metabolites as enhancers of venous thrombogenicity [81]. The AhR also plays a role in acute ischemic brain injury (stroke) induced by middle cerebral artery occlusion, and is accompanied increased levels of kynurenine in the brain. Considering that the AhR antagonist 2’,4’,6-trimethoxyflavone or inhibition of kynurenine synthesis decreased stroke-associated damage, it is conceivable that the AhR plays a role in this condition [82]. Several studies demonstrate the role of the AhR and its ligands (e.g., TCDD and benzo[a]pyrene) on bone, although there are some response-specific differences reported in rodent models [83,84,85,86,87]. In AhR null mice, there is an increase in bone volume fraction and decreased bone turnover and effects of ligands are variable and age-dependent. Loss of the AhR induces choroidal neovascular (CNV) lesion formation, and in a mouse model of this lesion two AhR-active pharmaceuticals (leflunomide and flutamide) attenuated CNV pathogenesis [88,89]. In addition to the effects of various AhR ligands on the multiple disease models noted above, the AhR and its ligands influence multiple diseases by modulating the immune system, and this has been extensively reviewed [18,22,48,49,50,51,52,53,70,72,73]. Thus, the AhR plays a role in cellular homeostasis and in pathophysiology (Figure 1), and like the NRs the AhR is a multi-tissue drug target.

## 4. Basis for Development of Selective Receptor Modulators (SRM)

Development of drugs that target intracellular receptors such as the AhR and NRs must take into account several important considerations that determine the potency and efficacy of a receptor ligand. In addition to absorption, distribution, metabolism and excretion (ADME), the agonist or antagonist activity of individual receptor ligands has been extensively investigated for steroid hormone receptors [90,91]. Figure 2 illustrates that the activity of the liganded AhR-complex is dependent on several factors. Interaction of ligands with a receptor induces ligand-specific conformational changes in the receptor-ligand complex, which in turn interacts with a tissue-specific set of nuclear cofactors, including coactivators, coregulators or corepressors. The ligand will also influence binding to cis-elements and posttranslational modifications of the receptor. All these factors combined are determinants in the tissue/cell-specific agonist or antagonist activity of the ligand. The classical examples of SRMs are the selective estrogen receptor modulators (SERMs) tamoxifen and raloxifene, which are extensively used for treatment of ER-positive breast cancer and loss of bone density, respectively [91,92]. Tamoxifen acts as an ER antagonist in mammary tumors but exhibits ER agonist activity in bone and the uterus; raloxifene is also an ER antagonist in mammary tumors and an ER agonist in bone, but is not an ER agonist in the uterus. These differences between tamoxifen and raloxifene in the uterus have been linked to the inability of the raloxifene–ER complex to recruit the coactivator SRC-1 in uterine tissues [93]. Thus, although SRMs are an important class of drugs, their receptor-mediated expression of genes and functions as agonists or antagonists is not readily predictable due to the multiple factors that regulate their activities (Figure 1). Thus, the activity of an SRM as an agonist or antagonist must be confirmed in functional and genomic assays.

## 5. Classification of AhR Ligands—SahRMs or Not?

AhR ligands have been classified into several different categories based on structural similarities, toxicities, their origins (xenobiotic, endobiotic and cognate), interactions with wild-type and mutant AhRs, rates of metabolism, persistence and ligand-binding affinities. A possible underlying assumption for these classifications is that AhR ligands that fall within a particular group will have similar functional activities differing primarily in their relative potencies that will be dependent, in part, to ADME considerations. An alternative to these classifications is to consider that AhR ligands are selective AhR modulators (SAhRMs), and although different groups of SAhRMs may exhibit overlapping activities, their genomic and functional activities may differ and these differences are not readily predicted. The following discussion will examine different groups of AhR ligands and demonstrate their SAhRM-like activities.

Persistent halogenated aromatics (DLCs) are SAhRMs. The AhR was first characterized as the intracellular receptor that bound and mediated the toxic effects caused by TCDD and related compounds [13,14]. Moreover, there was a correlation between the rank-order AhR binding affinities of TCDD and related planar halogenated aromatics and their potencies as inducers of CYP1A1. This correlation was also extended to other responses and led to the development of the toxic equivalency factor approach (TEF) for hazard assessment of these compounds [94,95,96]. It was assumed that for a mixture of toxic halogenated aromatics (HA’s), their dioxin-like equivalents (TEQ) were equal to the summation of the concentrations of their individual components’ times a TEF value. TEFs represent the fractional potency of a DLC compared to TCDD
TEQ = Σ[HA] × TEF(1)
which is assigned a value of 1.0. The assignment and refinement of TEFs for DLCs is periodically updated and takes into account new data [97,98,99]. The TEF approach has been used as guidance for regulatory agencies to promulgate rules to reduce generation and emission of halogenated aromatics into the environment. This has resulted in lower environmental levels and decreased accumulation of these persistent toxicants in fish, wildlife and humans. Inspection of the data for the 29 most environmentally relevant PCDD, PCDF and PCB congeners shows that for some compounds, their TEFs for different AhR-mediated responses can vary by 10- to 100-fold depending on the species and the responses. Since most of these compounds are not readily metabolized, this variability in potency relative to TCDD may be consistent with SAhRM-like activity. For example, genomic studies on analysis of gene expression data induced by DLCs in different tissues and species suggest that dioxin-like compounds (DLCs) are SAhRMs (rev in [100]). TCDD, 3,3′,4,4′,5-pentachlorobiphenyl (PCB126) and 2,3,7,8-tetrachlorodibenzofuran (TCDF) are potent DLCs. In mouse liver, these compounds induced changes in expression of 3280, 2343 and 1411 genes, respectively, and the number of overlapping genes for all three compounds was only 202 [101]. Some of these differences may be due to compound-specific induction of AhR-independent genes. A similar comparison of the effects of TCDD and TCDF in mouse liver showed that 1027 and 837 genes, respectively, were induced but only 373 gene were induced by both compounds [102]. These data coupled with results of other studies on genomic differences of DLCs demonstrate their unique effects on gene expression [100]. This suggests that DLCs are SAhRMs, and these genome level differences may explain some of the variability in their TEF values since their “selectivity” in gene expression may influence tissue-specific potency differences of DLCs.

### Non-DLCs as SAhRMs

The number of individual compounds that bind to and modulate AhR-mediated responses and genes is continually increasing, and includes structurally diverse synthetic chemicals (e.g., pharmaceuticals), phytochemicals, microbial metabolites and endogenous biochemicals, some of which may be “cognate” AhR ligands [4,5,6,94,96,103,104,105,106,107]. Most but not all of these compounds bind the receptor, with much lower binding affinities than observed for TCDD. Some of these AhR ligands fit into the category of rapidly metabolized AhR ligands (RMAhRLs) [106,107]. This part of the review will examine and provide some examples of how different classes of AhR ligands are highly selective in terms of their agonist and antagonist activities, suggesting that they are SAhRMs. A comprehensive screening of 596 pharmaceuticals as inducers of CYP1A1 in several organs identified a subset of 147 compounds that were further investigated for their activity in gel shift, reporter gene, receptor binding and CYP1A1 mRNA induction assays [108]. The results clearly showed compound-specific selectivity for this panel of AhR-responsive assays; for example, 81 compounds (59%) that induced mouse hepatic CYP1A1 did not bind or activate the AhR in vitro [108]. They identified only nine pharmaceuticals that activated the panel of AhR-dependent responses and did not induce dioxin-like activity. These compounds included nimodipine, leflunomide, flutamide, omeprazole, mexiletine and atorvastatin. Subsequent studies with these same compounds demonstrate the assay selectivity of these pharmaceuticals as AhR agonists or antagonists. For example, Figure 3 shows that mexiletine induces CYP1A1 mRNA in MDA-MB-468 (minimal) and BT47 breast cancer cells (Figure 3A); however, in these same cell lines mexiletine exhibits partial AhR antagonist activity and inhibits TCDD-induced CYP1A1 mRNA in MDA-MB-468 cells (Figure 3B). In contrast, omeprazole is a relatively potent AhR agonist (like TCDD) for induction of CYP1A1 in both cell lines (Figure 3C) and in MDA-MB-468 cell migration assays TCDD, and omeprazole but not mexiletine inhibits migration [109]. Results of other studies in breast and pancreatic cancers cells further demonstrate that the AhR activities of pharmaceuticals are highly selective [109,110].

Flavonoids are polyphenolic health-promoting phytochemicals in fruits and vegetables. A recent study in Caco2 colon cancer cells investigated the effects of structure, and response specificity of flavonoids as AhR ligands (Figure 4) [111]. Structure-activity relationships for induction of CYP1A1 were primarily dependent on the number of hydroxyl groups with pentahydroxy compounds such as quercetin-inducing responses with a magnitude similar to that observed for TCDD. Hexahydroxy flavonoids were less active, and tetrahydroxy compounds, including luteolin and apigenin (Figure 4A), were inactive as inducers of CYP1A1 (Figure 4B). In contrast, the magnitude of the induction of the AhR-responsive UGT1A1 gene by TCDD, quercetin apigenin and luteolin was similar (Figure 4C). In Caco2-AhR knockout cells, induction of CYP1A1 or UGT1A1 was not observed for flavonoids or TCDD [111]. It was also observed that both apigenin and luteolin inhibited TCDD-induced CYP1A1 in Caco2 cells (Figure 4D), demonstrating that the AhR agonist or antagonist activities of flavonoids in a specific cell line were dependent on the compound structure and specific response.

Recent reviews on the role of the AhR and its ligands in cancer demonstrated that the receptor exhibits tumor-specific suppressive or promoter-like activities and AhR ligands that are dioxin-like, and RMAhRLs can enhance or inhibit tumor growth [62,63]. Response-specific SAhRM-like activity has also been observed for AhR-active tryptophan metabolites. For example, kynurenine induced clot weights in a mouse model for venous thrombogenicity [80], whereas in a model for idiopathic pneumonia syndrome kynurenine inhibited lung damage [79] and the AhR antagonist CH223191 inhibited kynurenine-induced responses. AhR ligand-specific effects on LPS-induced genes in bone-marrow-derived macrophages were observed [111]; TCDD and FICZ but not I3C suppressed CCL20 gene expression and TCDD and FICZ induced IL-6 and IL-10, but expression of these cytokines was suppressed by indole-3-carbinol (I3C). The venous thrombogenicity and lung inflammation model and the bone-marrow-derived macrophages have an immune cell component, and there is extensive evidence in other disease models that involve inflammation and immune cells that different classes of AhR ligands selectively modulate AhR-mediated responses and genes [48,49,50,51,52,53].

The intestinal tract and its resiliency are dependent on the AhR and its ligands, which are derived from dietary sources and microbial metabolites. Expression of the AhR protects against intestinal inflammation and maintains barrier function. A recent review summarized several studies on the effects of structurally diverse AhR ligand on dextran sodium sulfate (DSS) and 2,4,6-trinitrobenzenesulfonic acid (TNBS)-induced intestinal inflammation in mouse models [112]. Some studies suggest differences between DLCs and RMAhRLs [99,100]; however, chemical-induced intestinal inflammation in mice is ameliorated by almost every class of AhR ligands, including DLCs (TCDD), synthetic (β-naphthoflavone), plant-derived (norisoboldine, broccoli, extracts, indole-3-carbinol, flavonoids and indigo extracts), microbial (DHNA, urothilin A, tryptophan diet) and endogenous (ITE and FICZ) compounds [19,47,113,114,115,116,117,118,119,120] (Figure 5). In contrast, oxazolone enhanced intestinal inflammation, and this effect is also AhR-dependent [121]. Moreover, there is evidence that the yeast malassezi retructa, which produces several AhR active compounds and enhances gut inflammation [122,123] and serotonin (an AhR ligand) [124], also enhanced inflammation; however, in these studies the role of the AhR was not investigated [125,126]. Thus, even for an organ that is protected by the AhR, there are some examples of selective AhR ligand-dependent responses and this is consistent with their SAhRM-like activity.

## 6. Summary

There is evidence that AhR ligands are highly selective in their activity and exhibit agonist or antagonist activities and potency differences that are tissue/organ/species-specific. This SAhRM-like activity can be observed in different structural and functional classes of AhR ligands, and may hold true even for high affinity ligands such as TCDD and other DLCs. For example, in pancreatic cancer cells the AhR-active pharmaceuticals omeprazole and tranilast but not TCDD inhibited invasion of Panc1 pancreatic cancer cells, and this response was AhR-dependent. The fact that AhR ligands are SAhRMs provides remarkable flexibility for the development of SAhRMs for clinical applications in treating multiple diseases. A major caveat or concern in the development of SAhRMs or ligands for any receptor is that their response selectivity as agonist or antagonists cannot be readily predicted and broad-based testing is necessary for identifying the optimal ligand for specific clinical applications.

## Figures and Tables

**Figure 1 ijms-21-06654-f001:**
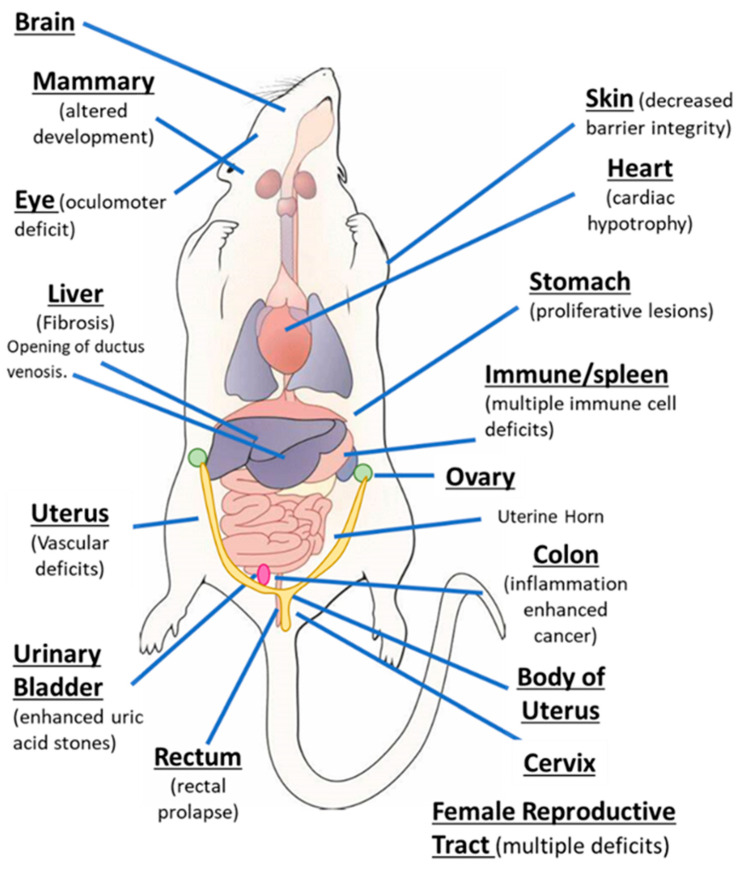
The aryl hydrocarbon receptor (AhR) plays a role in cellular homeostasis in several tissues/organs in the mouse models.

**Figure 2 ijms-21-06654-f002:**
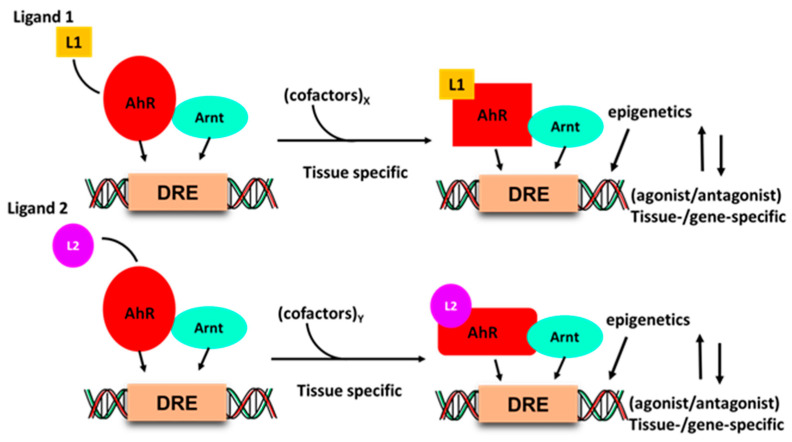
Ligand–AhR binding induces conformational changes in the AhR and AhR–Arnt complex, which differentially interacts with nuclear cofactors (coactivators/corepressors) in a tissue and gene-specific manner, and this may be accompanied by changes in chromatin and receptor-DNA binding.

**Figure 3 ijms-21-06654-f003:**
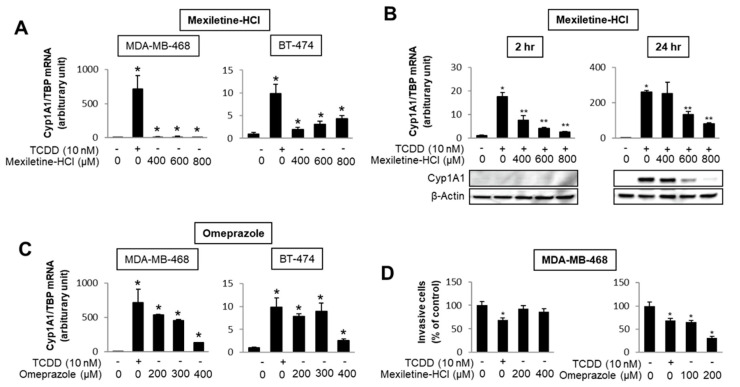
AhR-active pharmaceuticals as selective AHR modulators (SAhRMs). In MDA-MB-468 and BT474 breast cancer cells, mexiletine differentially induces CYP1A1 (**A**) but inhibits 2,3,7,8-tetrachlorodibenzo-p-dioxin (TCDD)-induced CYP1A1 mRNA and protein in MDA-MB-468 cells (**B**). Omeprazole is a potent inducer of CYP1A1 in both MDA-MB-468 and BT-474 cells (**C**) and TCDD and omeprazole, but not mexiletine-inhibited MDA-MB-468 cell invasion (**D**) [109]. Significant (*p* < 0.05) induction (*) and inhibition (**) is indicated.

**Figure 4 ijms-21-06654-f004:**
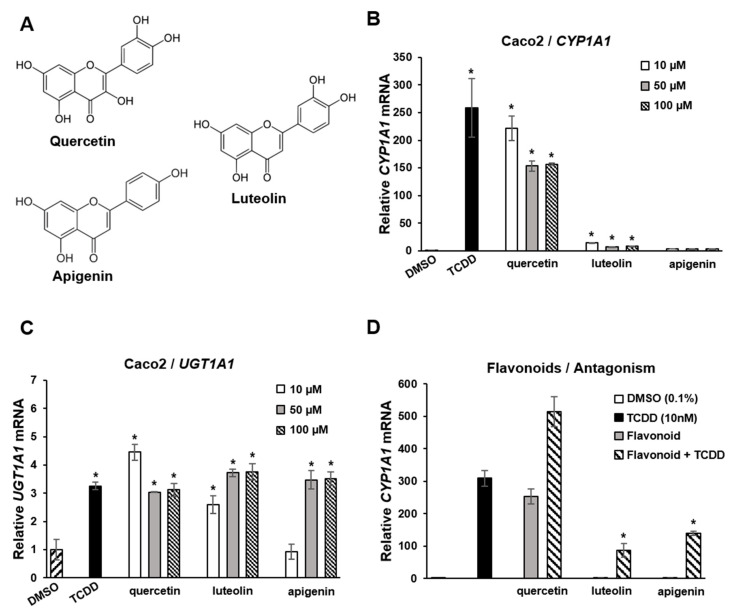
Flavoniods as SAhRMs. (**A**) Flavonoid structure. (**B**) TCDD and quercetin-induced CYP1A1 in Caco2 cells, luteolin and apigenin exhibited minimal to non-detectable activity, respectively. TCDD, quercetin, luteolin and apigenin induced UGT1A1 (**C**) and luteolin and apigenin, but not quercetin-inhibited induction of CYP1A1 by TCDD in Caco 2 cells (**D**) [111]. Significant (*p* < 0.05) induction (**B**,**C**) or inhibition of TCDD-induced CYP1A1 (**D**) is indicated (*).

**Figure 5 ijms-21-06654-f005:**
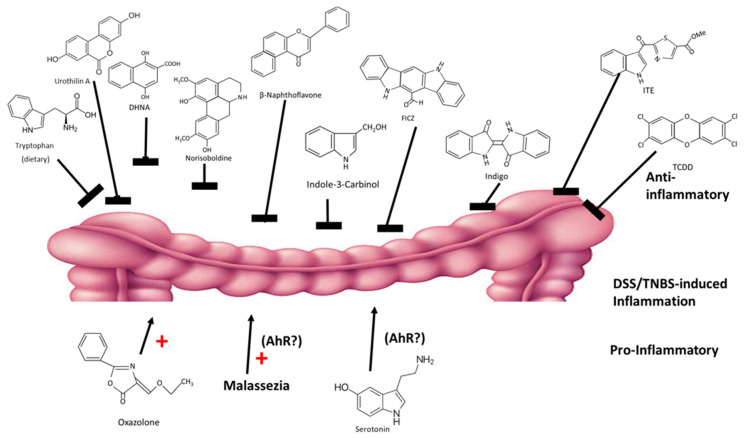
AhR ligands exhibit anti-inflammatory (top) and pro-inflammatory effects in mouse models of intestinal inflammation (rev. in [112]).

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
