# Peer review of "Aryl Hydrocarbon Receptor (AHR) Ligands as Selective AHR Modulators (SAhRMs)"

_ijms, 2020, doi:10.3390/ijms21186654_

Round 1

Reviewer 1 Report

This review titled, “Aryl hydrocarbon receptor ligands as selective AHR modulators” by Safe et al. is a good, informative review on the current knowledge of AHR ligands. This review article is well-written and should be very interesting to AHR scientists. It describes good summary of AHR function using AHR null mice. I have summarized some specific comments below.

  1. Not sure if abbreviations should be included in the title.
  2. Should delete the first paragraph under Introduction.
  3. Line 55, type: “2,3,7,8”
  4. lines 60-62, should include kynurenine as an endogenous ligand.
  5. Fig. 1 is hard to read. Need better image.
  6. Lines 92-94, “The advantages of targeting…with antagonists or agonists respectively” – should rewrite this sentence.
  7. Lines 96-105, should include effect on more causes such as breast, liver, lung where AHR is known to express and play some role.
  8. Lines 111-112, should rewrite; how does imiquimod-induced skin inflammation relate to AHR function?
  9. Line 147, ref 90, 91 not appropriate. Should use reference specific to AHR.
  10. Fig. 2, “Ligand 2” missing. Should link the above reference to Fig. 2.
  11. Lines 193-204, what about AHR independent gene expression as a possible explanation?
  12. Fig. 3, hard to read. Need bigger font size.
  13. Fig. 5, need better image on structures and structure names.
  14. Lines 282-285, should consider including results/hypotheses from Denison’s and Perdew’s groups to explain the observed differences in the ligand responsiveness of AHR.

Author Response

Reviewer 1

  1. I have left in the Abbreviation since it is recognizable by many scientists in the field.
  2. First paragraph has been deleted.
  3. “2,3,7,8”-has been corrected.
  4. Kynurenine has been added (lines 60-62)
  5. Figure 1 has been improved.
  6. Lines 92-94 have been rewritten.
  7. I have only expanded this section slightly (lines 96-105) since there is considerable variability on the role of the AhR using different models of the same tumor. The role of the AhR in head, neck and colon tumors are the same in all studies
  8. Lines 111 and 112 have now been rewritten.
  9. The section has been reworded and clarified (line 47).
  10. The figure has been corrected.
  11. Differences may also be due to activation of AhR-independent genes (lines 193-204) and this now noted.
  12. Figure 3 has now been enlarged.
  13. Figure 5 has been improved.
  14. There are now so many models for distinguishing between different classes of AhR ligands that we did not want to expand on any specific model (i.e.: Denison, Perdew, Kerkvliet, Bradfield and others).

Reviewer 2 Report

In this interesting review, the authors provide evidence supporting the hypothesis that AhR ligands are Selective AHR Modulators (SAhRMs) as their agonist/antagonist action function is specific and dependent on cell and tissue context. This assumption is relevant,  because it sustains the hypothesis of using selective SAhRMs in the therapy of diseases involving AhR, as inflammatory ones. 

I suggest to improve this job by considering:

  • Too long sentences, where the use of the conjunction "and" is too frequent while sometimes logical linkers are lacking. For example (lines 258-261): "Expression of the AhR protects against intestinal
    inflammation and maintains barrier function and several studies have investigated the effects of structurally diverse AhR ligand on dextran sodium sulfate (DSS) and 2,4,6-trinitrobenzenesulfonic
    261 acid (TNBS)-induced intestinal inflammation and this was summarized in a recent review (112)" could be rephrased as: "Expression of the AhR protects against intestinal
    inflammation and maintains barrier function. Several studies have investigated the effects of structurally diverse AhR ligand on dextran sodium sulfate (DSS) and 2,4,6-trinitrobenzenesulfonic acid (TNBS)-induced intestinal inflammation. This argument was summarized in a recent review (112)." Please, check these sentences.  
  • Sometimes, conclusions preceed data. i.e.: "The AhR also plays a role in acute ischemic brain injury (stroke) induced by middle cerebral artery occlusion and this was accompanied increased levels of kynurenine in the brain and the AhR antagonist 2´,4´,6-trimethoxyflavone (TMF) or inhibition of kynurenine synthesis decreased stroke-associated damage (82)." could be rephrased as: "Acute ischemic brain injury (stroke) induced by middle cerebral artery occlusion is accompanied by increased levels of kynurenine in the brain. Considering that the AhR antagonist 2´,4´,6-trimethoxyflavone (TMF) or inhibition of kynurenine synthesis decreased stroke-associated damage, it is conceivable that AhR also plays a role in this condition (82).
  • Please check for abbreviations near extended names and vice-versa.
  • lines 55-56: please, change 2,37,8-tetrachlorodibenzo-p-dioxin (TCDD) in 2,3,7,8-tetrachlorodibenzo-p-dioxin (TCDD).
  • line 92: please, delete repeated "that".
  • lines 102-105: please, break the sentence.
  • lines 111-112: please check the sentence where probably the effect of FICZ is not described.
  • line 118: please change "role" in "roles" and "study" in: different studies.
  • lines 121-124 and 128-132: please, break the sentences.
  • line 134: please, specify the effects of which ligands.
  • line 137. please, delete the comma after addition.
  • lines 149-152, 177-180, 186-189, 195-198, 252-256, 258-261, 261-266, : please,break the sentences.
  • line 195: please, delete the extra parenthesis.
  • line 246: please add "for" between "observed" and  "AhR-active".

Author Response

Reviewer 2 made several suggested changes in sentence structure (too long) and phraseology and we have tried to address his/her concerns in the revised manuscript, which has now been extensively edited.